# Lectin Analysis of SARS-CoV-2-Positive Nasopharyngeal Samples Using GLYcoPROFILE^®^ Technology Platform

**DOI:** 10.3390/diagnostics12112860

**Published:** 2022-11-18

**Authors:** Mateja Seničar, Benoît Roubinet, Richard Daniellou, Thierry Prazuck, Ludovic Landemarre

**Affiliations:** 1GLYcoDiag, 2 Rue du Cristal, 45100 Orléans, France; 2AgroParisTech, Cosmetology Department, 45100 Orléans, France; 3Centre Hospitalier Régional d’Orléans, Servicedes Maladies Infectieuses et Tropicales, 45100 Orléans, France

**Keywords:** SARS-CoV-2, COVID-19, nasopharyngeal samples, lectins, GLYcoPROFILE^®^ technology

## Abstract

Nasopharyngeal samples are currently accepted as the standard diagnostic samples for nucleic acid amplification testing and antigenic testing for the SARS-CoV-2 virus. In addition to the diagnostic capacity of SARS-CoV-2-positive crude nasopharyngeal samples, their qualitative potential for direct glycan-specific analysis, in order to uncover unique glycol profiles, was assessed. In this study we provide glycan characterization of SARS-CoV-2-positive and -negative nasopharyngeal samples directly from lectin interactions. Although with limited throughput, this study evaluated the clinical sensitivity and specificity of the GLYcoPROFILE^®^ technology platformon45crude nasopharyngeal samples collected between November 2020 and April 2022. Each GLYcoPROFILE^®^ of 39 SARS-CoV-2-positive samples was compared toglycoprofiling on a panel of 10 selected lectins and the results were paralleled with SARS-CoV-2-negative samples’ results. The GLYcoPROFILE^®^ showed a clear distinction between positive and negative samples with WFA, GSL-II, PHA-L (Glc*N*Ac-specific) and BPA (Gal*N*Ac-specific) highlighted as relevant lectins in SARS-CoV-2-positive samples. In addition, a significant, positive statistical correlation was found for these lectins (*p* < 0.01).

## 1. Introduction

Coronaviruses (CoVs) are a large and diverse group of positive-stranded RNA viruses, which were discovered in the human population only six decades ago. In the 1960s, the first human coronavirus CoV-229E (HCoV-229E) was discovered and isolated from six medical students in Chicago [1,2,3,4]. Since then, a total of seven human coronaviruses have been identified, including HCoV-229E, HCoV-OC43, HCoV-HKU1, HCoV-NL63, MERS-CoV, SARS-CoV-1 and SARS-CoV-2 [1,2,3,4]. The most recent emergence of the novel coronavirus, SARS-CoV-2, which causes severe acute respiratory coronavirus disease 2019 (COVID-19), from Wuhan, Hubei Province, China, in December 2019, started the still-ongoing pandemic declared by the World Health Organization (WHO) in March 2020 [5,6,7,8,9,10].

Structurally, the SARS-CoV-2 virus contains an envelope that exhibits glycoproteins such as spike (S-protein) and membrane (M-protein) proteins. These proteins, especially S-proteins, have the central role in viral pathogenesis and attachment to the angiotensin-converting enzyme 2 (ACE2), mainly expressed in the host lung’s epithelial cells [11,12]. The presence of glycoproteins in the viral envelope of SARS-CoV-2 also opens a wide range of possibilities for the application of carbohydrate-binding agents in clinical analysis, such as lectins, that recognize specific types of carbohydrates residues [13,14]. The S-protein is heavily glycosylated with both 3 *O*-glycosylation and 22 *N*-glycosylationsites, mainly carrying D-mannose, D-glucose, *N*-acetyl-D-galactosamineand *N*-acetyl-D-glucosamine glycan motifs, and more rarely D-fucose and D-rhamnose [11,15].

In the present work, we have probed the recognition potency of numerous lectins, with diverse but known sugar specificities, towards SARS-CoV-2-positive nasopharyngeal samples in order to: (i) further characterize SARS-CoV-2 glycans using our GLYcoPROFILE^®^ technology platform and (ii) consequently discriminate them from SARS-CoV-2-negative nasopharyngeal samples. To the best of our knowledge, this study constitutes the first in situ glycan assessment of crude nasopharyngeal samples, positive for SARS-CoV-2, in direct interaction with lectins (Figure 1).

## 2. Materials and Methods

### 2.1. General Remarks

Fluorescence was recorded with a FLUOstar^®^ OPTIMA spectrometer (BMG Labtech, Offenburg, Germany). LEctPROFILE^®^ plates were obtained from GLYcoDiag (Orléans, France). Nasopharyngeal swabs were collected by trained personnel using flexible nylonflocked swabs (Ningbo Dasky Life Sciences Co., Ltd., Ningbo, China) at the Centre Hospitalier Régional d’Orléans (CHRO) from hospitalized patients for COVID-19 and were conserved in sampling tubes containing sterile physiological saline solution (3 mL, 0.85% NaCl/H_2_O) at −20 °C until the GLYcoPROFILE^®^ analysis. The studied population includes 45 participants aged ≥ 18 years who were tested for SARS-CoV-2 between November 2020 and April 2022 at the CHRO. The heat map data were analyzed using the GraphPad Prism Version 5.03 for Windows (GraphPad Software, San Diego, CA, USA).The RT-PCR test for SARS-CoV-2 was performed in the virology unit of the Centre Hospitalier Régional d’Orléans (CHRO), France. Nucleic acid extraction was performed with an automated sample preparation system MGISP-960 (MGI, Shenzhen, China). Real-time PCR detection of SARS-CoV-2 RNA targeting the ORF1ab, S and N genes was performed with the TaqPath V2 COVID-19 Multiplex RT-PCR kit (Thermo Fisher, Illkirch, France). Amplification was performed on QuantStudio^TM^5 (Applied Biosystems^TM^, Singapore). The assay includes an internal RNA extraction control and an amplification control. The assay was performed according to the manufacturer’s instructions. The samples were analyzed taking into account the new positivity criteria of the French Microbiology Society’s (FMS) expert committee (Version 4 of 14 January 2021), in particular, taking into account the specific characteristics of the Thermo Fisher kit used for the RT-PCR measurement.

### 2.2. Antibodies and Reagents

All laboratory chemicals used, including the starting compounds, reagents and solvents, were analytical-grade purity and commercially available, unless otherwise specified. Primary and secondary antibodies: anti-SARS-CoV-2 spike S1 protein azide-free antibody (Diaclone, Besançon, France), biotin-SP (long spacer) AffiniPure F(ab′)₂ fragment goat anti-human IgG (H + L) polyclonal antibody (Jackson Immuno Research Labs, West Grove, PA, USA); other reagents: fluorescein (DTAF) streptavidin (Jackson Immuno Research Labs, West Grove, PA, USA).

### 2.3. GLYcoPROFILE^®^ of Nasopharyngeal Samples

Lectin array assays were performed according to GLYcoDiag’s protocol, as already described [14,15,16,17]. Briefly, non-diluted crude nasopharyngeal samples (50 μL each) were deposited in each well of lectin in triplicate and incubated for 1h at room temperature. After washing with 0.05%Tween20/PBS buffer, the anti-SARS-CoV-2 antibody was added (50 μL, 1/100) and incubated for 30 min. Then, the plates were washed with 0.05%Tween20/BS buffer and the secondary antibody, goat anti-human IgG, was added (50 μL, 1/3000) and incubated for another 30 min, washed and fluorescein (DTAF) streptavidin (100 μL, 1/600) was added. After 30min of incubation, the plate was finally washed with 0.05%Tween20/PBS and 100 μL of PBS was further added before plate reading by a fluorescence reader (λ_ex_ = 485 nm, λ_em_ = 530 nm), FLUOstar^®^ OPTIMA spectrometer (BMG Labtech, Offenburg, Germany). The intensity of the signal is directly correlated with the ability of the virus to be recognized by the lectin. Table 1, below, summarizes the specificity of the lectins used in this study.

## 3. Results

### 3.1. GLYcoPROFILE^®^—Lectin-Based Glycoprofiling of SARS-CoV-2-Positive Nasopharyngeal Samples

For the initial lectin-based glycoprofiling of crude SARS-CoV-2-positive nasopharyngeal samples, a panel of 20 different lectins (from natural and bacterial sources) with broad sugar specificities was used in order to visualize the general type and tendencies of glycosylation (Table 1). The performance of the assay was examined on 4 SARS-CoV-2-positive nasopharyngeal samples collected during the period between November 2020 and March 2021. As this glycoprofiling assay was intended for use as a complementary test following a positive RT-PCR detection of SARS-CoV-2, evaluation of negative samples was not performed. From the characterized samples, 10 lectins (HHA, BPA, WFA, MPA, AIA, GSL-II, WFA, PHA-L, PHA-E, SNA) that provided fluorescence signal intensities above the threshold of 4000 and in addition had more narrow glycan specificity than the counterpart lectins, were selected as a relevant group for the glycoprofiling of SARS-CoV-2 samples (Figure 2)

Then, using these 10 selected lectins, the GLYcoPROFILE^®^ study (Figure 3) was extended to a larger panel including 45 participants of whom 39 tested positive for SARS-CoV-2 and 6 were negative(both positive and negative tests were confirmed by antigenic testing). Among the positive group of samples, based on the fluorescence intensity data represented in the histogram, the higher interaction is apparent on all screened lectins except HHA lectin, which shows visibly lower interactions. The six negative samples generally showed low interactions with all the lectins, with a fluorescence intensities threshold value of around 1000. Through the interpretation of the GLYcoPROFILE^®^ raw data in Figure 3, the lectin HHA can be highlighted as a negative reference lectin for evaluation of SARS-CoV-2-positive samples, whilst for the rest of the lectins, the intensity interactions are ambiguous, and interpreted in this graphical representation.

### 3.2. Interpretation of GLYcoPROFILE^®^—Discrimination between SARS-CoV-2-Positive and -Negative Crude Nasopharyngeal Samples

To further highlight the GLYcoPROFILE^®^ differences of SARS-CoV-2-positive (Group A) and -negative samples (Group B) and to put in evidence specific lectin interactions, the heat map representation of average fluorescence values was used and reported in Figure 4. As depicted, two lectins, BPA and PHA-E, were able to discriminate the control group from SARS-CoV-2-positive samples, with values around or higher than 3000. The BPA recognizes Gal*N*Ac-containing glycans, while PHA-E binds to complex Gal*N*Ac glycans. In addition, another group of lectins, including WFA, GSL-II and PHA-L, specific for Glc*N*Ac-containing glycans, were distinguishing for SARS-CoV-2-positive samples. The HHA lectin was confirmed as a negative reference lectin as firstly indicated by Figure 3.

Given the simplicity of the visualization of the heat map data representation, a more detailed insight into each individual GLYcoPROFILE^®^ of SARS-CoV-2-positive and -negative sample is reported in Figure 5. The fluorescence intensities of each glycoprofiled sample is represented according to heat map color gradient code, with positive, higher interaction values indicated with warmer colors, and negative, lower interaction values indicated with colder colors. This representation allows us to clearly underline the general higher interaction tendencies of the SARS-CoV-2-positive samples (group A), Figure 5A, towards selected lectins and to distinguish them from the SARS-CoV-2-negative samples (group B). In addition, viral charge data were included to possibly put in perspective these higher interaction tendencies observed for the group A samples. The viral charges are based on the RT-PCR amplification of gene coding for the N nucleocapsid protein of the SARS-CoV-2 virus and the interpretation of values is defined by French Microbiology Society’s expert committee (Version 4 of 14 January 2021), where the high viral charge is <25, moderate is 25–30 and low is >30. The viral charge data were available only for the 21 samples positive for SARS-CoV-2 virus included in this study; therefore, they were used only subsequently as supplementary data in order to correlate their GLYcoPROFILE^®^ with viral charge. These samples were classified according to their increasing viral charges, from moderate to high Figure 5B. An overall evaluation of these samples points to a connection between high viral charge and higher lectin interaction pattern, mostly with the previously highlighted lectins, BPA, WGA, GSL-II, PHA-L and PHA-E. In addition, the statistical analysis *t*-test was applied to confirm these observations Table 2. Based on the interpretation of *p*-value (Table 2), there are three lectins (WFA, SNA and AIA) that are statistically significant with the *p*-value range of 0.01 ≤ *p* < 0.05 and four lectins (GSL-II, WGA, BPA and PHA-L) that are in the *p*-value range of 0.001 ≤ *p* < 0.01, which characterizes them as highly statistically significant. This indeed validates that this group of lectins can correctly discriminate positive SARS-CoV-2 samples from negative SARS-CoV-2 samples within GLYcoPROFILE^®^ detection technology.

## 4. Discussions

The coronavirus SARS-CoV-2 has been, for the past two years, the focus of the scientific community as well as the general public, and its virology, epidemiology, etiology, diagnosis and treatment are extensively well documented [8]. The scientific studies that focused on the glycolprofiling of the SARS-CoV-2 virus were mostly in vitro and did not employ the lectin-based techniques or the crude nasopharyngeal samples from the positive patients [12]. Herein, we wanted to investigate the in situ lectin–glycan interactions of viral glycoproteins and broaden the use of GLYcoPROFILE^®^ technology that has already been well established for several biotechnological applications [14,15,16,17]. Our main goal was to find a group of lectins that will distinguish SARS-CoV-2-positive and -negative samples based on their GLYcoPROFILE^®^. After preliminary screening study of a few SARS-CoV-2-positive samples on a large panel of 20 lectins with complementary and superposed glycan-recognition specificities, we selected a group of 10 lectins that showed higher interaction tendencies. From the average of GLYcoPROFILEs^®^, we were able to clearly distinguish SARS-CoV-2-positive samples from negative samples with the lectins WFA, GSL-II, BPA and PHA-L as good discriminants. It is interesting to note that these four lectins are specific for Glc*N*Ac (WFA, GSL-II, PHA-L)- and Gal*N*Ac (BPA)-containing glycans, which are abundant in SARS-CoV-2 glycoproteins. Although there are some limitations in this study as the sample size of SARS-CoV-2-negative samples was relatively scarce, mostly due to their limited availability during the early stages of the pandemic, we validated GLYcoPROFILE^®^ technology as being capable of distinguishing SARS-CoV-2-positive and -negative samples.The obtained GLYcoPROFILE^®^ results are encouraging to further investigate the qualitative potential of the direct lectin-based glycoprofiling of biologically complex crude samples. Analysis of a larger number of SARS-CoV-2-negative samples can further contribute to the validation of these preliminary results. Moreover, the GLYcoPROFILE^®^ of other human respiratory viruses, such as seasonal influenza, should be analyzed to exclude the cross-reactivity of glycan profiles that can coincide with SARS-CoV-2 and influenza virus coinfections [18,19].

## 5. Conclusions

Here, we addressed the performance analysis of GLYcoPROFILE^®^ technology, a lectin-based glycoprofiling platform, by using complex biological SARS-CoV-2-positive nasopharyngeal samples. The GLYcoPROFILE^®^ technology clearly indicated the glycan differences between SARS-CoV-2-positive and -negative samples, which are in accordance with bibliographically available data. Even though this short study was exploratory towards the glycoprofiling of biologically more complex samples, it shows the versatility of GLYcoPROFILE^®^ technology and its potential for adaptation for a broader application in diagnostic activities.

## Figures and Tables

**Figure 1 diagnostics-12-02860-f001:**
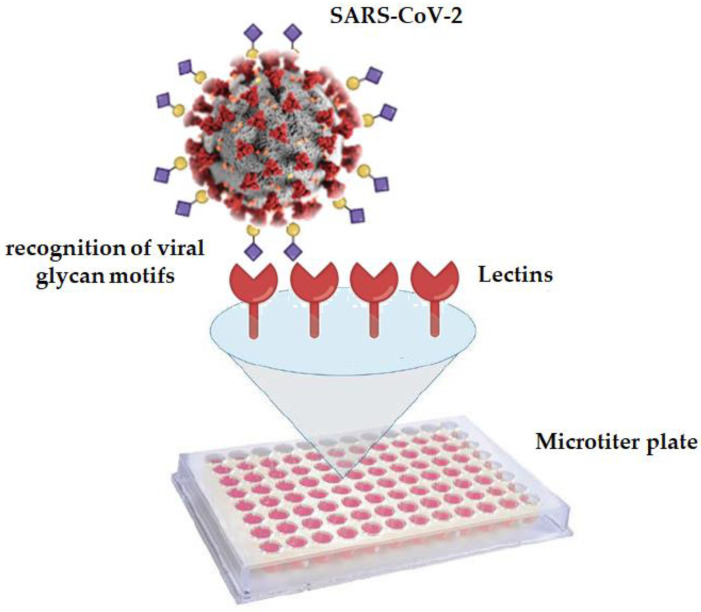
Schematic representation of GLYcoPROFILE^®^ technology. Recognition of SARS-CoV-2 glycan motifs by specific lectins immobilized on the microtiter plate.

**Figure 2 diagnostics-12-02860-f002:**
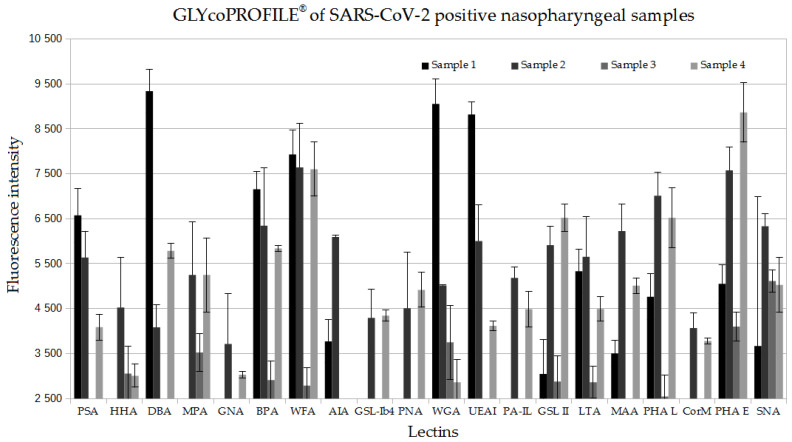
GLYcoPROFILE^®^ raw data of 4 SARS-CoV-2-positive nasopharyngeal samples on a panel of 20 lectins. Non-diluted samples were screened against selected lectins, previously adsorbed on LEctPROFILE^®^ plate, and revealed with anti-SARS-CoV-2 antibodies (1/100).

**Figure 3 diagnostics-12-02860-f003:**
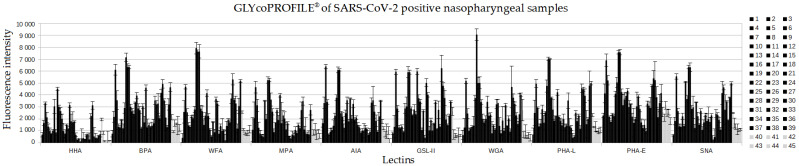
GLYcoPROFILE^®^ raw data of SARS-CoV-2-positive (N°1–39 in black) and -negative (N°40–45 in grey) nasopharyngeal samples on a panel of 10 lectins. Non-diluted samples were screened against selected lectins, previously adsorbed on LEctPROFILE^®^ plate, and revealed with anti-SARS-CoV-2 antibodies (1/100).

**Figure 4 diagnostics-12-02860-f004:**
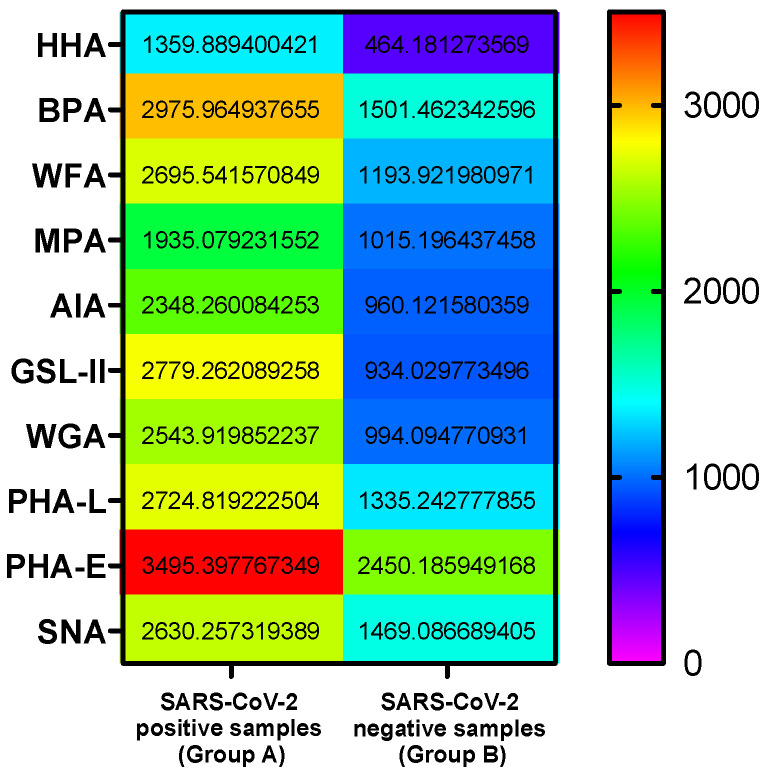
Average of GLYcoPROFILE^®^ fluorescence intensities interactions of nasopharyngeal SARS-CoV-2-positive and -negative samples with lectins.

**Figure 5 diagnostics-12-02860-f005:**
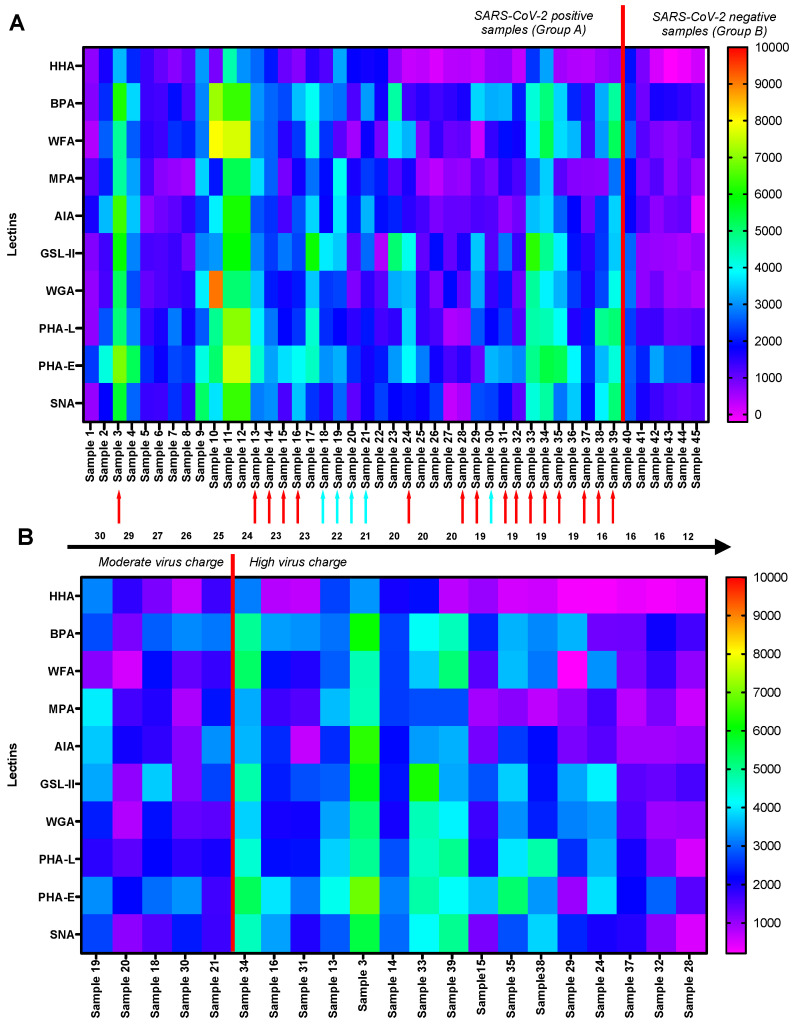
GLYcoPROFILE^®^ Heat Map representation of the fluorescence intensities of raw data of individual SARS-CoV-2-positive (from N°1 to 39) and -negative samples (from N°40 to 45). (**A**) GLYcoPROFILE^®^ Heat Map with indicated viral charges (red arrow: high viral load, blue arrow: moderate viral load, no arrow: viral load not determined); (**B**) Regrouped GLYcoPROFILE^®^ Heat Map according to increasing (moderate to high) viral charge. Interpretation of viral charge results for N gene amplification of SARS-CoV-2 virus according to FMS: high < 25, moderate 25–30, low > 30.

**Table 1 diagnostics-12-02860-t001:** Specificity of lectins used for GLYcoPROFILE^®^ analysis of SARS-CoV-2-positive and -negative nasopharyngeal samples.

Lectin Abbreviation	Common Name	Glycan Specificity
PSA	*Pisum sativum agglutinin*	Manα6(Manα3)Man
HHA	*Hippeastrum hybrid agglutinin*	terminal and internal Man
DBA	*Dolichos biflorus agglutinin*	αGal*N*Ac; terminal Gal*N*Ac; Gal*N*Acα3Gal*N*Ac
MPA	*Maclurapomifera agglutinin*	Galβ3Gal*N*Ac (T antigen), Galα6Glc (melibiose)
GNA	*Galanthus nivalis agglutinin*	Manα6(Manα3)Man
BPA	*Bauhinia purpurea agglutinin*	Galβ3Gal*N*Ac (T-antigen), Gal*N*Ac, Gal
WFA	*Wisteria floribunda agglutinin*	Gal*N*Acα6Gal, Gal*N*Acα3Gal*N*Ac, Gal*N*Ac
AIA	*Artocarpus intergrifolia agglutinin*	Galβ3Gal*N*Ac
GSL-Ib4	*Griffoniasimplicifolia isoB4*	αGal
PNA	*Arachis hypogaea agglutinin*	Galβ3Gal*N*Ac
WGA	*Triticum vulgare agglutinin*	(Glc*N*Acβ4)n, Neu5Ac; poly(*N*-acetyllactosamine)
UEA-I	*Ulex europaeus*	Glc*N*Acβ4Glc*N*Ac oligomers; Fucα2Galβ4Glc*N*Ac
PA-IL	*Pseudomonas aeruginosa lectin A (Lec A)*	terminal αGal
GSL II	*Griffonia simplicifolia lectin*	α or βGlc*N*Ac; αgalactosylated tri/tetra antennary glycans; core 3 *O*-glycans
LTA	*Lotus tetragonolobus agglutinin*	Fuc, LeX but not LeA, 2-FucαGalβ4Glc
MAA	*Maackiaamurensis agglutinin*	Neu5Acα2, 3Gal4Gal*N*Ac
PHA-L	*Phaseolus vulgaris agglutinin*	Complex glycans
PHA-E	*Phaseolus vulgaris agglutinin*	Complex glycans
CorM	*Coregonus lavaretusmarenae*	Rha
SNA	*Sambucus nigra agglutinin*	Neu5Acα2,6Gal/Gal*N*Ac

**Table 2 diagnostics-12-02860-t002:** Statistical analysis *t*-test for SARS-CoV-2-positive and -negative samples. Only the positive samples with high viral charge < 25 were taken for the calculation. The *p*-value under 0.05 can be considered as significant.

Lectins	*p*-Value
GSL-II	0.002
WGA	0.005
BPA	0.007
PHA-L	0.007
WFA	0.024
SNA	0.033
AIA	0.048

## Data Availability

The datasets used and/or analyzed during the current study are available from the corresponding authors on reasonable request.

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
