# Peer review of "Lectin Analysis of SARS-CoV-2-Positive Nasopharyngeal Samples Using GLYcoPROFILE® Technology Platform"

_diagnostics, 2022, doi:10.3390/diagnostics12112860_

Round 1

Reviewer 1 Report

The chosen subject is of great interest. I think that the Abstract part can be improved a little and be clearer.

1. I did not find in the templates where to refer to the Abstract. 2. The topic is relevant and interesting in the context of the pandemic which is not over yet. 3. The article brings a new approach to the method 4. The methodology is comprehensive. 5. The conclusions are based on the presented results. 6. References are good. 7. I recommend modifying the Abstract in order to make it clearer and more understandable.

Author Response

Changes have been made in the Abstract (sentences have been added).

Abstract: Nasopharyngeal samples are currently accepted as the standard diagnostic samples for the nucleic acid amplification testing and antigenic testing for the SARS-CoV-2 virus. In addition to the diagnostic capacity of SARS-CoV-2 positive crude nasopharyngeal samples, their qualitative potential for direct glycan-specific analysis, in order to uncover unique glycoprofile, was assessed. In this study we provide glycan characterization of SARS-CoV-2 positive and negative nasopharyngeal samples directly from lectin interactions. Although with limited throughput, this study evaluated the clinical sensitivity and specificity of GLYcoPROFILE® technology platform on 45 crude nasopharyngeal samples collected between November 2020 and April 2022. Each GLYcoPROFILE® of 39 SARS-CoV-2 positive samples was confronted to glycoprofiling on a panel of 10 selected lectins and the results were paralleled with SARS-CoV-2 negative samples results. The GLYcoPROFILE® showed the clear distinction between positive and negative samples with WFA, GSL-II, PHA-L (GlcNAc specific) and BPA (GalNAc specific) highlighted as relevant lectins for SARS-CoV-2 positive samples. In addition, a significant positive statistical correlation was found for these lectins (p < 0.01).

Reviewer 2 Report

In the current research article by Senicar et al., the authors have tried to characterize the glycans on SARS-CoV2 based on their specific interactions with different lectins in GLYcoPROFILE assay. For this, they have used crude nasopharyngeal samples of SARS-CoV2 positive (39) and negative patients (6). Their ultimate goal was to identify specific interactions that can differentiate SARS CoV2 positive and negative samples to explore the potential of this assay/technology in diagnostics. Their study identified 4 lectin/glycan interactions significantly appearing in CoV2+ samples. While their results are encouraging and interesting there are some potential limitations, one of them being a small size of CoV2- samples that the authors have already mentioned. To address this issue, they could enroll some healthy volunteers to collect the samples. The biggest risk of trusting this data is owing to significantly less number of negative samples. This is because with the data provided, it is difficult to judge the cross reactivity of SARS-Cov2 with other common viruses in terms of these glycan interactions. Since most of the mammalian viruses have their envelope proteins heavily glycosylated, there is no supporting data to prove the glycan specificities of SARS CoV2 in light of other viruses. Hence, in the clinical setting how will such an assay be able to reliably detect SARS CoV2 positive samples is very questionable. I agree that the authors mentioned, the study is exploratory in nature and is actually far from clinical application without having some key experiments performed to assess the technology. Another interesting thing to investigate how emerging mutants of CoV2 would affect the glycan interactions as viruses often escape the host immune pressure by losing/gaining glycan moeities. With all these concerns, I think the study could be improved further. Overall, I appreciate the use complex biological samples in the study and the data obtained based on them. It would be important to explain some of these limitations in the discussion section with additional references.

Author Response

Changes have been made in the Discussion (sentences have been added to address the limitations of the study and corresponding literature references [18,19]).

The coronavirus SARS-CoV-2 has been for the past two years in the focus of scientific community as well as general public and its virology, epidemiology, etiology, diagnosis, and treatment are extensively well-documented[8]. The scientific studies that focused on the glycoprofiling of the SARS-CoV-2 were mostly in vitro and did not employ the lectin-based techniques or the crude nasopharyngeal samples from the positive patients[12]. Herein, we wanted to investigate the in situ lectin–glycan interactions of viral glycoproteins and broaden the use of GLYcoPROFILE® technology that has already been well-established for several biotechnological applications[14-17]. Our main goal was to find a group of lectins that will distinguish SARS-CoV-2 positive and negative samples based on their GLYcoPROFILE®. After preliminary screening study of few SARS-CoV-2 positive samples on large panel of 20 lectins with complementary and superposed glycan recognition specificities, we selected a group of 10 lectins that showed higher interactions tendencies. From the average of GLYcoPROFILEs® we were able to clearly distinguish SARS-CoV-2 positive from negative samples with the lectins WFA, GSL-II, BPA and PHA-L as good discriminants. It is interesting to note that these four lectins are specific for GlcNAc (WFA, GSL-II, PHA-L) and GalNAc (BPA) containing glycans which are abundant in SARS-CoV-2 glycoproteins. Although there are some limitations in this study as sample size of SARS-CoV-2 negative samples was relatively scarce, mostly due to their limited availability during the early stages of the pandemic, we validated GLYcoPROFILE® technology as being capable of distinguishing SARS-CoV-2 positive and negative samples. The obtained GLYcoPROFILE® results are encouraging to further investigate the qualitative potential of direct lectin-based glycoprofiling of biologically complex crude samples. Analysis of a larger number of SARS-CoV-2 negative samples can further contribute to validation of this preliminary results. Moreover, the GLYcoPROFILE® of other human respiratory viruses, such as seasonal influenza, should be analysed to exclude the cross-reactivity of glycan profiles that can coincide with SARS-CoV-2 and influenza virus coinfections[18,19].

18. Z. Abdelrahman, M. Li, X. Wang, Comparative Review of SARS-CoV-2, SARS-CoV, MERS-CoV, and Influenza A Respiratory Viruses. Front. Immunol., 2020, 11, 552909. doi.org/10.3389/fimmu.2020.552909

19. L. Bai, Y. Zhao, J. Dong, S. Liang, M. Guo, X. Liu, X. Wang, Z. Huang, X. Sun, Z. Zhang, L. Dong, Q. Liu, Y. Zheng, D. Niu, M. Xiang, K. Song, J. Ye, W. Zheng, Z. Tang, M. Tang, Y. Zhou, C. Shen, M. Dai, L. Zhou, Y. Chen, H. Yan, K. Lan, K. Xu, Coinfection with influenza A virus enhances SARS-CoV-2 infectivity. Cell Research, 2021, 31, 395-403. doi.org/10.1038/s41422-021-00473-1

Reviewer 3 Report

This is a well written short communication

Few comments

1. limitations should be addressed

2. The p‐value under 0,05?? shouldnt it be 0.05? a lot of areas are using coma instead of period. Please correct

3. Table 2 is not required. Just need to list the significant ones. Not required to show all the p values

4. Several grammatical errors need to corrected. English language correction is required

5. Discussion needs to be more elaborate. Currently is very short

Author Response

1. limitations should be addressed

Changes have been made in the Discussion (sentences have been added to address the limitations of the study and corresponding literature references [18,19]).

The coronavirus SARS-CoV-2 has been for the past two years in the focus of scientific community as well as general public and its virology, epidemiology, etiology, diagnosis, and treatment are extensively well-documented[8]. The scientific studies that focused on the glycoprofiling of the SARS-CoV-2 were mostly in vitro and did not employ the lectin-based techniques or the crude nasopharyngeal samples from the positive patients[12]. Herein, we wanted to investigate the in situ lectin–glycan interactions of viral glycoproteins and broaden the use of GLYcoPROFILE® technology that has already been well-established for several biotechnological applications[14-17]. Our main goal was to find a group of lectins that will distinguish SARS-CoV-2 positive and negative samples based on their GLYcoPROFILE®. After preliminary screening study of few SARS-CoV-2 positive samples on large panel of 20 lectins with complementary and superposed glycan recognition specificities, we selected a group of 10 lectins that showed higher interactions tendencies. From the average of GLYcoPROFILEs® we were able to clearly distinguish SARS-CoV-2 positive from negative samples with the lectins WFA, GSL-II, BPA and PHA-L as good discriminants. It is interesting to note that these four lectins are specific for GlcNAc (WFA, GSL-II, PHA-L) and GalNAc (BPA) containing glycans which are abundant in SARS-CoV-2 glycoproteins. Although there are some limitations in this study as sample size of SARS-CoV-2 negative samples was relatively scarce, mostly due to their limited availability during the early stages of the pandemic, we validated GLYcoPROFILE® technology as being capable of distinguishing SARS-CoV-2 positive and negative samples. The obtained GLYcoPROFILE® results are encouraging to further investigate the qualitative potential of direct lectin-based glycoprofiling of biologically complex crude samples. Analysis of a larger number of SARS-CoV-2 negative samples can further contribute to validation of this preliminary results. Moreover, the GLYcoPROFILE® of other human respiratory viruses, such as seasonal influenza, should be analysed to exclude the cross-reactivity of glycan profiles that can coincide with SARS-CoV-2 and influenza virus coinfections[18,19].

18. Z. Abdelrahman, M. Li, X. Wang, Comparative Review of SARS-CoV-2, SARS-CoV, MERS-CoV, and Influenza A Respiratory Viruses. Front. Immunol., 2020, 11, 552909. doi.org/10.3389/fimmu.2020.552909

19. L. Bai, Y. Zhao, J. Dong, S. Liang, M. Guo, X. Liu, X. Wang, Z. Huang, X. Sun, Z. Zhang, L. Dong, Q. Liu, Y. Zheng, D. Niu, M. Xiang, K. Song, J. Ye, W. Zheng, Z. Tang, M. Tang, Y. Zhou, C. Shen, M. Dai, L. Zhou, Y. Chen, H. Yan, K. Lan, K. Xu, Coinfection with influenza A virus enhances SARS-CoV-2 infectivity. Cell Research, 2021, 31, 395-403. doi.org/10.1038/s41422-021-00473-1

2. The p‐value under 0,05?? Shouldn’t it be 0.05? a lot of areas are using comma instead of period. Please correct

Changes have been made. The comma was replaced by period as decimal separator.

Table 2. Statistical analysis T-test for SARS-CoV-2 positive and negative samples. Only the positive samples with high viral charge <25 were taken for the calculation. The p-value under 0.05 can be considered as significant.

3. Table 2 is not required. Just need to list the significant ones. Not required to show all the p values

Changes have been made. Only the p-values for significant lectins have been kept in the Table 2.

4. Several grammatical errors need to corrected. English language correction is required

Grammatical errors and English language corrections have been made.

5. Discussion needs to be more elaborate. Currently is very short

Changes have been made in the Discussion (sentences have been added to elaborate it and corresponding literature references [18,19]).

The coronavirus SARS-CoV-2 has been for the past two years in the focus of scientific community as well as general public and its virology, epidemiology, etiology, diagnosis, and treatment are extensively well-documented[8]. The scientific studies that focused on the glycoprofiling of the SARS-CoV-2 were mostly in vitro and did not employ the lectin-based techniques or the crude nasopharyngeal samples from the positive patients[12]. Herein, we wanted to investigate the in situ lectin–glycan interactions of viral glycoproteins and broaden the use of GLYcoPROFILE® technology that has already been well-established for several biotechnological applications[14-17]. Our main goal was to find a group of lectins that will distinguish SARS-CoV-2 positive and negative samples based on their GLYcoPROFILE®. After preliminary screening study of few SARS-CoV-2 positive samples on large panel of 20 lectins with complementary and superposed glycan recognition specificities, we selected a group of 10 lectins that showed higher interactions tendencies. From the average of GLYcoPROFILEs® we were able to clearly distinguish SARS-CoV-2 positive from negative samples with the lectins WFA, GSL-II, BPA and PHA-L as good discriminants. It is interesting to note that these four lectins are specific for GlcNAc (WFA, GSL-II, PHA-L) and GalNAc (BPA) containing glycans which are abundant in SARS-CoV-2 glycoproteins. Although there are some limitations in this study as sample size of SARS-CoV-2 negative samples was relatively scarce, mostly due to their limited availability during the early stages of the pandemic, we validated GLYcoPROFILE® technology as being capable of distinguishing SARS-CoV-2 positive and negative samples. The obtained GLYcoPROFILE® results are encouraging to further investigate the qualitative potential of direct lectin-based glycoprofiling of biologically complex crude samples. Analysis of a larger number of SARS-CoV-2 negative samples can further contribute to validation of this preliminary results. Moreover, the GLYcoPROFILE® of other human respiratory viruses, such as seasonal influenza, should be analysed to exclude the cross-reactivity of glycan profiles that can coincide with SARS-CoV-2 and influenza virus coinfections[18,19].

18. Z. Abdelrahman, M. Li, X. Wang, Comparative Review of SARS-CoV-2, SARS-CoV, MERS-CoV, and Influenza A Respiratory Viruses. Front. Immunol., 2020, 11, 552909. doi.org/10.3389/fimmu.2020.552909

19. L. Bai, Y. Zhao, J. Dong, S. Liang, M. Guo, X. Liu, X. Wang, Z. Huang, X. Sun, Z. Zhang, L. Dong, Q. Liu, Y. Zheng, D. Niu, M. Xiang, K. Song, J. Ye, W. Zheng, Z. Tang, M. Tang, Y. Zhou, C. Shen, M. Dai, L. Zhou, Y. Chen, H. Yan, K. Lan, K. Xu, Coinfection with influenza A virus enhances SARS-CoV-2 infectivity. Cell Research, 2021, 31, 395-403. doi.org/10.1038/s41422-021-00473-1